# Recovery of Pd(II) from Aqueous Solution by Polyethylenimine-Crosslinked Chitin Biosorbent

Zhuo Wang [1] , Su Bin Kang [1] and Sung Wook Won [1,2,*]

1   Department of Ocean System Engineering, College of Marine Science, Gyeongsang National University, Tongyeong 53064, Korea; wz928661411@gmail.com (Z.W.); svkang123@gmail.com (S.B.K.)
2   Department of Marine Environmental Engineering, College of Marine Science, Gyeongsang National University, Tongyeong 53064, Korea
*   Correspondence: sungukw@gnu.ac.kr; Tel.: +82-55-772-9136

**Abstract:** This study reports the recovery of Pd(II) from acid solution by a polyethylenimine (PEI)-crosslinked chitin (PEI-chitin) biosorbent. FE-SEM analysis demonstrated that there are many slot-like pores on PEI-chitin. The $N_2$ adsorption–desorption experiment revealed that the average pore size was 47.12 nm. Elemental analysis verified the successful crosslinking of PEI with raw chitin. The Langmuir model better explained the isotherm experimental data and the theoretical maximum Pd(II) uptake was 57.1 mg/g. The adsorption kinetic data were better described by the pseudo-second-order model and the adsorption equilibrium was achieved within 30 min for all initial Pd(II) concentrations of 50–200 mg/L. In the fixed-bed column, the adsorption of Pd(II) on PEI-chitin showed a slow breakthrough and a fast saturation performance. The desorption experiments achieved a concentration factor of 8.4 ± 0.4; in addition, the adsorption–desorption cycles in the fixed-bed column were performed up to three times, consequently confirming the good reusability of PEI-chitin for Pd(II) recovery. Therefore, the PEI-chitin can be used as a promising biosorbent for the recovery of Pd(II) in practical applications.

**Keywords:** palladium; chitin; crosslinking; adsorption; reusability

## 1. Introduction

Palladium, one of the platinum group metals (PGMs), has been extensively applied in various fields, including catalysis, electronics, energy, pharmaceuticals, hydrogen storage, and jewelry due to its remarkable physicochemical properties [1,2]. The proven global reserves of palladium are only 67,000 tons, making it one of the most expensive resources [2]. The abundance of palladium in primary ores is extremely low (~0.01 g/kg), and extracting palladium from ores does not only consume large amounts of energy and water but also generates large quantities of wastes [3]. On the contrary, the content of palladium in end-of-life products, such as spent catalyst and waste printed circuit boards, is as high as 0.1–10 g/kg [3,4]; therefore, recovery of palladium from end-of-life products is of great significance, including extending the life of raw ores, saving energy and water, and reducing waste generation [5].

Different methods such as adsorption [6], solvent extraction [7], precipitation [8], as well as biological method [9] have been developed for palladium recovery However, most of them have many shortcomings, such as high investment, low efficiency, and use of a lot of chemicals [10]. In the above methods, adsorption is regarded as a simple, inexpensive, and cost-effective option because of its easy operation, low cost, and reuse of adsorbents [11]. However, conventional adsorbents such as activated carbons are quite expensive and not environmentally friendly [12]. In this respect, today's researchers are focusing on the use of biological wastes to develop adsorbents [13], which can not only convert biological wastes into valuable things, but also solve the pollution problems caused by them.

Chitin is a renewable, inexpensive, and non-toxic biological material which exists in large quantities in seafood wastes [14]; furthermore, chitin is rich in N-acetyl and hydroxyl groups, making it a potential adsorbent [15]. Moreover, chitin has good stability against strong acids, so it is of great significance for the recovery of PGMs [16] because aqua regia is widely used for leaching PGMs from ores and urban mines [17,18]; however, the direct use of chitin as an adsorbent for PGMs recovery has rarely been studied, which may be attributed to its very low adsorption capacity [19]. Therefore, in the past few decades, researchers have focused on applying deacetylated chitin, namely chitosan, as an adsorbent for PGMs recovery [20–22]. Nevertheless, the deacetylation of chitin not only requires a large volume of extremely high concentration NaOH solution but also produces a large amount of strongly alkaline wastewater [23]; moreover, since chitosan is water-soluble at pH < 6 [24], it is not suitable for use in recovering PGMs leached with strong acids. On the other hand, because chitin has enough amine and hydroxyl groups, it can be modified to improve its adsorption performance; thus, if the weakness of chitin can be compensated, promising acid-resistant biosorbents based on chitin can be developed to recover PGMs from acidic media.

Recently, researchers have put forward several approaches to enhance the adsorption performance of chitin in wastewater treatment. Cao et al. (2018) prepared a porous chitin adsorbent by gel method for the treatment of methylene blue wastewater, and its adsorption capacity was increased about 13 times [25]. Li et al. (2019) improved the adsorption capacity of chitin to indium (III) by 1.5 times through ultrasound modification [26]. Mamah et al. (2020) fabricated a palygorskite–chitin hybrid nanomaterial by a ball milling method, and its adsorption capacity for Pb(II) was increased from 6.3 to 53.7 mg/g [27]. In addition to the above achievements, our research team also successfully improved the adsorption capacity of chitin for Acid Blue 25 and Reactive Black 5 by crosslinking polyethylenimine (PEI) on the chitin surface [28,29]. Considering that there is insufficient research on the recovery of PGMs using chitin-based biosorbents, it is of great significance to investigate the adsorption performance of the PEI-crosslinked chitin (PEI-chitin) biosorbent for PGMs.

In this study, we developed a PEI-chitin biosorbent to recover Pd(II) via our previously reported method [28]. A field emission scanning electron microscope (FE-SEM), a Brunauer–Emmett–Teller (BET) analyzer, and an element analyzer were applied to characterize the raw chitin and PEI-chitin. Batch adsorption experiments, such as adsorption isotherms and adsorption kinetics, were carried out to evaluate the adsorption performance of Pd(II) on PEI-chitin. Batch desorption studies were performed to explore the optimal eluent concentration to desorb Pd(II) from the exhausted PEI-chitin. In addition, the reusability of PEI-chitin for Pd(II) recovery was estimated in fixed-bed columns.

## 2. Materials and Methods

### 2.1. Materials

The chitin powder, with a particle size of 180–300 μm, was prepared using the chitin flakes provided by Young Puk Chemical Co., Ltd. (Sokcho, Korea). PEI ($M_w$ = 70,000, 50% solution) was supplied by Habjung Moolsan Co., Ltd. (Seoul, Korea). Glutaraldehyde (GA, 25% solution) was obtained from Junsei Chemical Co., Ltd. (Tokyo, Japan). $PdCl_2$ with 99.0% purity was bought from Kojima Chemicals Co., Ltd. (Saitama, Japan). Palladium standard solution (1000 μg/mL) was supplied by PerkinElmer Inc. (Singapore). Hydrochloric acid (extra pure, above 35%) was supplied by Daejung Chemicals & Metals Co., Ltd. (Siheung, Korea).

### 2.2. Preparation of PEI-Chitin

The PEI-chitin was prepared using our previous method with slight modification [28]. Briefly, 3 g of chitin powder was mixed with 100 mL of 3% GA solution and stirred in an incubator under 25 °C and 160 rpm. After 2 h, the supernatant was removed, and then the remaining GA-treated chitin was washed 3 times with deionized water to eliminate residual GA. The GA-treated chitin was then mixed with 100 mL of 10 wt% PEI solution

for 30 min under the same conditions; thereafter, the unreacted PEI was removed and the residual PEI on the PEI-chitin surface was cleared away by deionized water. Finally, the PEI-chitin was dried in a freeze-dryer (HC-4110, HyperCOOL, LaboGene, Daejeon, Korea) for 24 h and then stored in a desiccator for further use.

### 2.3. Characterization Analysis

The surface morphologies of raw chitin and PEI-chitin were obtained using an FE-SEM equipped with an energy dispersive spectrometer (EDS) (JSM-7610F, JEOL, Akishima, Japan). The specific surface area, pore-size distribution, and pore volume of raw chitin and PEI-chitin were determined via $N_2$ adsorption–desorption isotherms using a specific surface area analyzer (3Flex, Micromeritics, Norcross, GA, USA). The elements' (C, N, O, and S) composition of raw chitin and PEI-chitin were measured using a micro automated elemental analyzer (TruSpec Micro, LECO, Anyang, Korea).

### 2.4. Batch Adsorption Studies

Before the experiments, a certain amount of $PdCl_2$ was dissolved in a 0.1 M HCl solution to prepare 1000 mg/L of Pd(II) stock solution. All working solutions were diluted from the stock solution using 0.1 M HCl. Batch adsorption experiments were conducted in 50 mL falcon tubes under 25 °C. In all experiments, the amount of biosorbent and volume of solution was 0.03 g and 30 mL, respectively. Pd(II) solutions, with the initial concentration ranging from 0 to 250 mg/L, were used for the isotherm experiments. The isotherm experiments were remained sufficiently for 24 h to confirm the adsorption equilibrium. Then, samples were gathered from the final solutions for concentration measurement. Kinetic experiments were carried out at initial Pd(II) concentrations of 50, 100, and 200 mg/L, respectively, and these concentrations were in the range of real Pd(II)-containing wastewater. Samples were collected from the supernatant regularly, then centrifuged at 10,000 rpm for 10 min, and diluted with distilled water if necessary. Hereafter, the Pd(II) concentration of the samples was measured using an inductively coupled plasma optical emission spectroscopy (ICP-OES, Avio 200, PerkinElmer, Singapore). The Pd(II) uptake $q$ (mg/g) was calculated using Equation (1):

$$q = \frac{C_i V_i - C_f V_f}{m} \tag{1}$$

where $C_i$ and $C_f$ (mg/L) are the initial and final concentrations of Pd(II), respectively. $V_i$ and $V_f$ (mL) are the initial and final volumes of the Pd(II) solution, and $m$ (g) is the mass of the adsorbent.

### 2.5. Batch Desorption Studies

To determine the optimal eluent concentration, various acidified thiourea solutions (1–10 mmol/L thiourea in 0.01 mol/L HCl solution) were used in the desorption experiments. First of all, 0.03 g of PEI-chitin was mixed with 30 mL of Pd(II) solution (100 mg/L) at 25 °C and 160 rpm for 2 h. Then, the Pd(II)-loaded PEI-chitin was rapidly rinsed with 0.01 M HCl solution twice. Hereafter, acidified thiourea solutions were used to desorb Pd(II) from exhausted PEI-chitin. The samples collected from the adsorption and desorption processes were centrifuged at 10,000 rpm for 10 min and diluted properly, and the residual Pd(II) concentrations were measured by ICP-OES. The desorption efficiency of PEI-chitin was calculated using Equation (2):

$$\text{Desorption efficiency (\%)} = \frac{\text{Desrobed Pd(II) amount (mg)}}{\text{Adsorbed Pd(II) amount (mg)}} \times 100\% \tag{2}$$

### 2.6. Reuse Studies

Dynamic adsorption–desorption experiments were carried out in a glass column (7 mm inner diameter and 10 cm length) at 26.0 ± 1.0 °C. Both ends of the fixed-bed column were equipped with absorbent cotton to avoid adsorbent leakage. Before each adsorption

and desorption process, the column was eluted with 0.01 M HCl solution for 30 min. The bed height and flow rate were kept at 5 cm (equal to 0.38 g PEI-chitin) and 1 mL/min, respectively. In the adsorption cycle, the initial Pd(II) concentration was 50 mg/L, the adsorption process was maintained for 360 min, and samples were collected at regular time intervals. The desorption experiment lasted for 90 min, the most suitable eluent determined in the batch desorption study was used to desorb Pd(II) from the adsorbents, and samples were gathered at predetermined times. The adsorption–desorption cycle was repeated 3 times to confirm the reusability of PEI-chitin in a fixed-bed column system. After each cycle of adsorption and desorption, the sample was centrifuged at 10,000 rpm for 10 min, diluted appropriately as needed, and then the Pd(II) concentration was determined by ICP-OES. The basic equations for column experimental data calculation are given below:

The treated solution volume, $V_t$ (mL), can be calculated using Equation (3):

$$V_t = Q \times t_{total} \tag{3}$$

where $Q$ (mL/min) is the flow rate, and $t_{total}$ (min) is the column working time.

The adsorption amount of Pd(II) at equilibrium $m_{eq}$ (mg) in the column is expressed using Equation (4):

$$m_{eq} = \frac{QA}{1000} = \frac{Q}{1000} \int_{t=0}^{t=t_{total}} (C_i - C_t)\mathrm{d}t \tag{4}$$

where $A$ represents the area below the breakthrough curve (plotted by $(C_i - C_t)$ vs. $t$), $C_i$ and $C_t$ (mg/L) are the Pd(II) concentrations in influent and effluent, respectively.

The adsorbed Pd(II) $q_{eq}$ (mg/g) at equilibrium is given by Equation (5):

$$q_{eq} = \frac{m_{eq}}{M} \tag{5}$$

where $M$ (g) is the dry weight of adsorbent in the column.

The total amount of Pd(II) ions in influent, $m_{total}$ (mg), is calculated by Equation (6):

$$m_{total} = \frac{QC_i t_{total}}{1000} \tag{6}$$

The removal rate of Pd(II) ions, $R$, is depicted by Equation (7):

$$R\ (\%) = \frac{m_{eq}}{m_{total}} \times 100 \tag{7}$$

The breakthrough time ($t_b$) and exhaustion time ($t_e$) is defined as the Pd(II) concentration in effluent is 5% ($C_t/C_0 = 0.05$) and 95% ($C_t/C_0 = 0.95$) of the influent, respectively.

## 3. Results

### 3.1. Characterization of Raw Chitin and PEI-Chitin

The surface morphologies of raw chitin and PEI-chitin recorded by FE-SEM are displayed in Figure 1. Figure 1a,b are the images of raw chitin obtained at ×500 and ×5000 magnification, respectively. It can be seen that the raw chitin is a porous material with a layered structure. The images of PEI-chitin captured at ×500 and ×5000 magnification are displayed in Figure 1c,d, respectively. No apparent changes were observed after PEI crosslinking, indicating that surface modification treatment has little effect on the structure of the raw chitin. To confirm whether PEI was crosslinked with chitin, elemental analysis was performed on raw chitin and PEI-chitin. Table 1 shows the chemical compositions (C, N, O, and S) of raw chitin and PEI-chitin. The nitrogen content of PEI-chitin was 0.5% higher than that of the raw chitin, indicating a small amount of PEI was crosslinked to the chitin surface. To elucidate the effect of PEI crosslinking on the structure of chitin, the BET surface area, average pore size, and pore volume of raw chitin and PEI-chitin were evaluated by $N_2$ adsorption–desorption isotherms and the results are shown in Figure 2 and Table 1. As depicted in Figure 2, the isotherms of raw chitin and PEI-chitin were in

type II shape with type H3 hysteresis loops, indicating the existence of mesopores and macropores in raw chitin and PEI-chitin [30]. The pore distribution of raw chitin and PEI-chitin (insert figure in Figure 2) also confirmed the existence of mesopores and macropores, and the pore volume of raw chitin was greatly reduced after crosslinking with PEI. As listed in Table 2, the pore volume and BET surface area of raw chitin (0.029 cm$^3$/g and 4.24 m$^2$/g) were higher than that of the PEI-chitin (0.015 cm$^3$/g and 2.09 m$^2$/g), while the average pore size of raw chitin (39.82 nm) was smaller than that of the PEI-chitin (47.12 nm). These could be attributed to the smaller pores that were covered by PEI molecules, resulting in a decrease in pore volume and BET surface area, and an increase in average pore size. The large average pore size can ensure the Pd(II) ions quickly diffuse through or into the pores of PEI-chitin [31], thus reaching equilibrium in a short time.

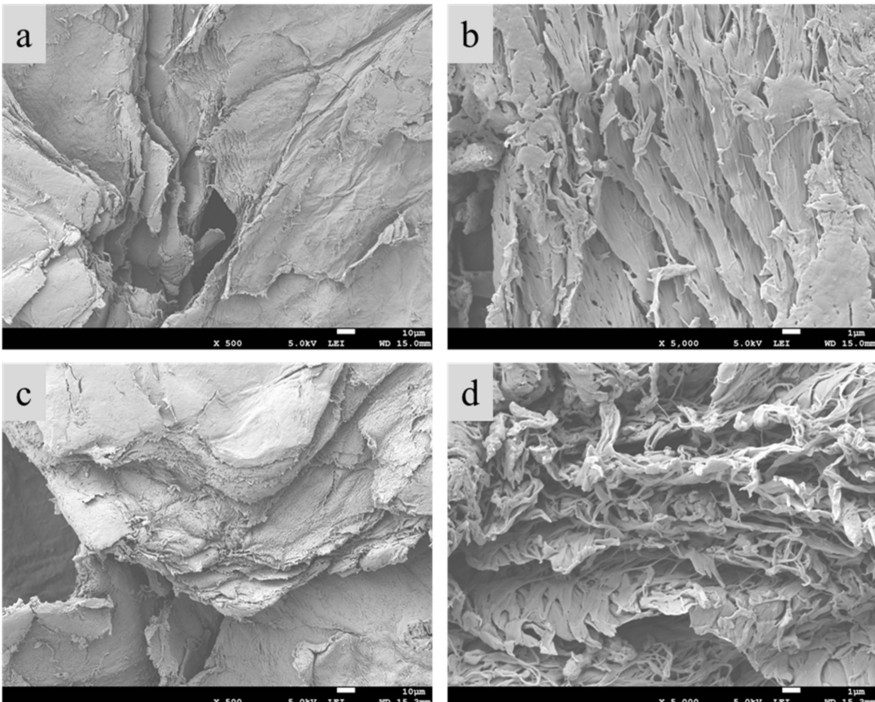

**Figure 1.** SEM images of (**a**,**b**) raw chitin and (**c**,**d**) PEI-chitin (**a**,**c**: ×500 magnification, and **b**,**d**: ×5000 magnification).

**Table 1.** Elemental analysis results of raw chitin and PEI-chitin.

| Sample | $M_{sample}$ | N (%) | C (%) | H (%) | S (%) |
|---|---|---|---|---|---|
| Raw chitin | 2.3350 | 5.45 | 42.55 | 4.36 | 0.97 |
| PEI-chitin | 2.1880 | 5.95 | 42.11 | 4.51 | 0.99 |

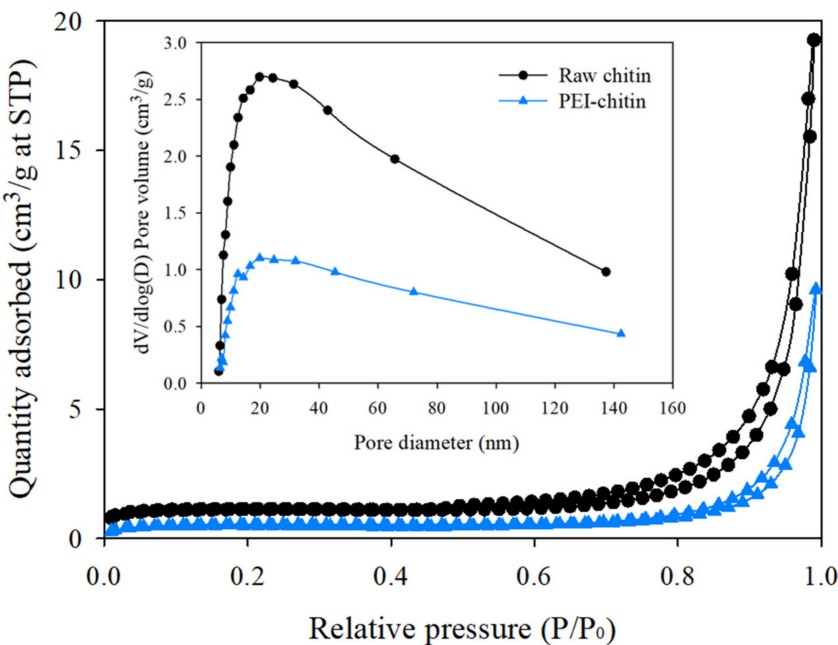

**Figure 2.** N$_2$ adsorption–desorption isotherms and pore size distributions (insert figure) of raw chitin and PEI-chitin.

**Table 2.** Pore volume, specific surface area, and average pore size of raw chitin and PEI-chitin.

| Sample | Pore Volume (cm$^3$/g) | BET Surface Area (m$^2$/g) | Average Pore Size (nm) |
|---|---|---|---|
| Raw chitin | 0.029 | 4.24 | 39.82 |
| PEI-chitin | 0.015 | 2.09 | 47.12 |

*3.2. Adsorption Isotherms*

Isotherm experiments were performed to assess the maximum Pd(II) uptake of raw chitin and PEI-chitin, and the results are displayed in Figure 3. The adsorption capacity of PEI-chitin for Pd(II) increased notably with the increase in Pd(II) concentration and then reached adsorption equilibrium at high Pd(II) concentration. By contrast, the Pd(II) uptake on raw chitin was negligible, hence we discarded it from the rest of the experiments. To clarify the adsorption characteristics, the Langmuir and Freundlich models were applied to depict the adsorption performance of Pd(II) on raw chitin and PEI-chitin. The equations of the Langmuir and Freundlich models are as follows:

$$\text{Langmuir model}: q_e = \frac{q_{max}K_LC_e}{1 + K_LC_e} \tag{8}$$

$$\text{Freundlich model}: q_e = K_FC_e^{1/n} \tag{9}$$

where $q_e$ (mg/g) is the Pd(II) uptake at equilibrium, $q_{max}$ (mg/g) is the maximum Pd(II) uptake, $C_e$ (mg/L) represents the Pd(II) concentration at equilibrium, $K_L$ (L/mg) and $K_F$ ((mg/g)(L/mg)1/$n$) are the Langmuir and Freundlich constants, respectively, and $n$ represents the Freundlich heterogeneity factor. The corresponding parameters fitted by the Langmuir and Freundlich models are listed in Table 3.

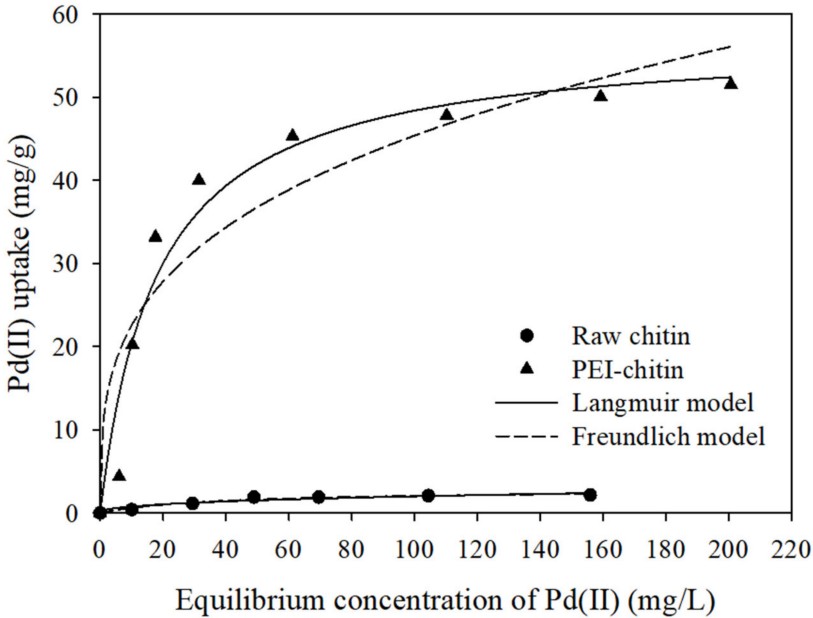

**Figure 3.** The adsorption isotherms of raw chitin and PEI-chitin for Pd(II) fitted by Langmuir and Freundlich models.

**Table 3.** Isotherm parameters of Pd(II) adsorption onto raw chitin and PEI-chitin.

| Sample | $q_{exp}$ (mg/g) | Langmuir Model | | | Freundlich Model | | |
|---|---|---|---|---|---|---|---|
| | | $q_{max}$ (mg/g) | $K_L$ (L/mg) | $R^2$ | $K_F$ ((mg/g)(L/mg)$^{1/n}$) | $n$ | $R^2$ |
| Raw chitin | 2.1 | 2.8 | 0.0275 | 0.9640 | 0.31 | 2.46 | 0.9083 |
| PEI-chitin | 51.5 | 57.1 | 0.0552 | 0.9541 | 11.19 | 3.29 | 0.8712 |

As displayed in Table 3, the coefficient of determination ($R^2$) of the Langmuir model was higher than that of the Freundlich model, indicating monolayer and homogeneous adsorption [32]. The maximum adsorption capacities of Pd(II) on raw chitin and PEI-chitin calculated by the Langmuir model were closer to their experimental results. The theoretical maximum adsorption capacity of Pd(II) on PEI-chitin (57.1 mg/g) was about 20 times higher than that of the raw chitin (2.8 mg/g). Furthermore, the $K_L$ value of PEI-chitin was higher than that of the raw chitin, demonstrating that the PEI-chitin has a higher affinity for Pd(II) [31], which was consistent with the experimental results. Overall, the adsorption of PEI-chitin for Pd(II) was favorable and the Langmuir model was more suitable for describing its adsorption performance.

### 3.3. Adsorption Kinetics

In practical applications, there is a great need for adsorbents with fast kinetics for adsorbates; therefore, the adsorption kinetics of Pd(II) by PEI-chitin were studied at the initial Pd(II) concentration of 50, 100, and 200 mg/L, respectively, and the results are shown in Figure 4. The PEI-chitin exhibited rapid adsorption kinetics for Pd(II) under the experimental conditions. The adsorption equilibrium was reached between 10 and 30 min at initial Pd(II) concentrations ranging from 50 to 200 mg/L. To elucidate the adsorption behavior, non-linear pseudo-first-order and pseudo-second-order models were used to describe the experimental results. The kinetic modes are given below:

$$\text{Pseudo-first-order model: } q_t = q_1(1 - \exp(-k_1 t)) \tag{10}$$

$$\text{Pseudo- sec ond-order-model: } q_t = \frac{q_2^2 k_2 t}{1 + q_2 k_2 t} \tag{11}$$

where $q_t$ (mg/g) is the Pd(II) uptake at time $t$ (min), $q_1$ and $q_2$ (mg/g) are the calculated amounts of Pd(II) adsorbed at equilibrium. $k_1$ (L/min) is the pseudo-first-order rate constant and $k_2$ (g/(mg·min)) is the pseudo-second-order rate constant.

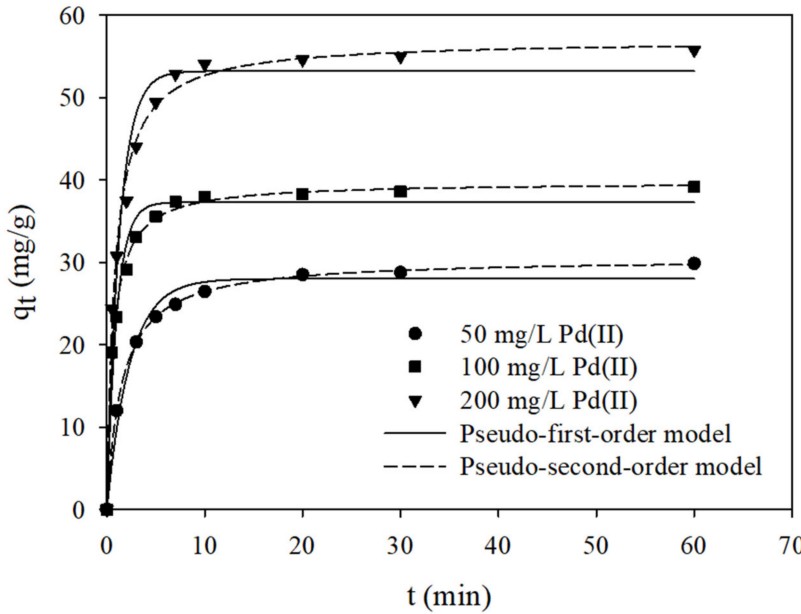

**Figure 4.** Pseudo-first-order and pseudo-second-order fitting of the adsorption kinetics of Pd(II) by PEI-chitin at the initial Pd(II) concentrations of 50, 100, and 200 mg/L.

Table 4 shows the parameters derived by the pseudo-first-order and pseudo-second-order models. The $R^2$ values of the pseudo-second-order model surpassed those of the pseudo-first-order model, and the equilibrium Pd(II) uptakes $q_2$ were closer to the experimental values. This indicated that the pseudo-second-order model was more suitable for explaining the adsorption kinetics of Pd(II) on PEI-chitin, and chemisorption was dominant in the adsorption process [33].

**Table 4.** Kinetic parameters of Pd(II) adsorption onto PEI-chitin at different initial Pd(II) concentrations.

| Pd(II) (mg/L) | $q_{exp}$ (mg/g) | Pseudo-First-Order Model | | | Pseudo-Second-Order Model | | |
|---|---|---|---|---|---|---|---|
| | | $q_1$ (mg/g) | $k_1$ (L/min) | $R^2$ | $q_2$ (mg/g) | $k_2$ (g/(mg·min)) | $R^2$ |
| 50 | 29.9 | 28.0 | 0.4268 | 0.9799 | 30.5 | 0.0214 | 0.9998 |
| 100 | 39.2 | 37.3 | 0.9999 | 0.9702 | 39.8 | 0.0403 | 0.9960 |
| 200 | 55.8 | 53.2 | 0.7784 | 0.9633 | 57.0 | 0.0216 | 0.9923 |

### 3.4. Desorption Studies

From the perspectives of Pd(II) recovery, value-added, and adsorbent recycling, the desorption of Pd(II) from the loaded adsorbent is a key factor for designing the adsorption process. In recent years, many researchers have used acidified thiourea solution as a high-efficient eluent to desorb Pd(II) from Pd(II)-loaded adsorbents [34]; therefore, we selected an acidified thiourea solution as an eluent for the desorption experiment. The acidified thiourea solutions (0–10 mmol/L) were prepared by adding a certain amount of thiourea in 0.01 mol/L HCl solution. The effect of thiourea concentration on the desorption efficiency of Pd(II) was evaluated, and the results are presented in Figure 5. With the thiourea concentration increased from 0 to 3 mmol/L, the desorption efficiency dramatically increased from 0% to 90.10%, then as the thiourea concentration continued to increase to 10 mmol/L, the desorption efficiency slowly increased to 99.88%. Therefore, a 10 mmol/L

acidified thiourea solution was determined as the most suitable eluent for the regeneration studies.

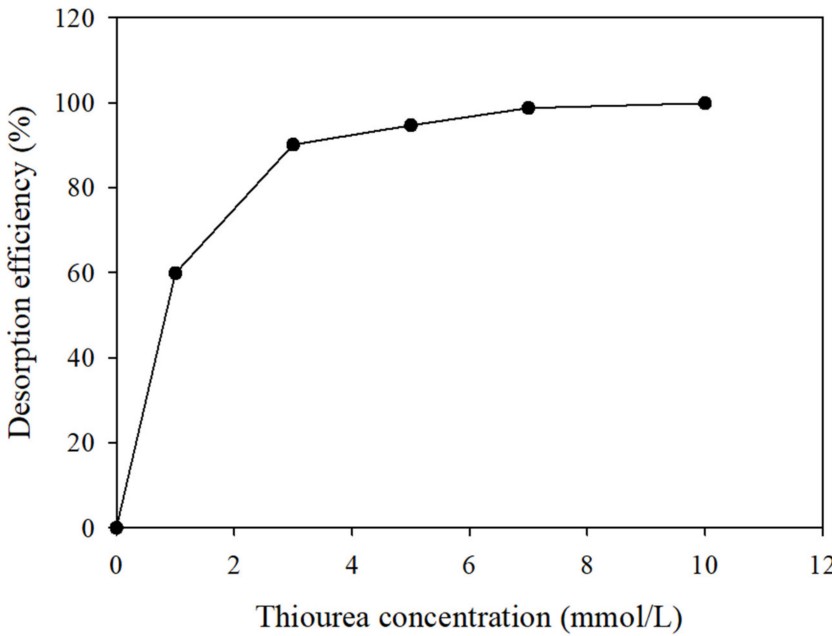

**Figure 5.** Pd(II) desorption using different concentrations of thiourea in 0.01 M HCl solutions.

*3.5. Reuse Studies in Fixed-Bed Column*

Good reusability is necessary for prolonging the service time of adsorbents and reducing the cost. Comparing to batch adsorption, fixed-bed column adsorption is more preferable and industrially feasible [35]; therefore, the regeneration experiment was conducted in a fixed-bed column to evaluate the reusability of PEI-chitin for Pd(II) recovery, and the results are demonstrated in Figure 6. As illustrated in Figure 6a, the PEI-chitin showed good stability in the adsorption process in a fixed-bed column, because no significant changes were observed on the breakthrough curves. The adsorption capacity of PEI-chitin for Pd(II) was 32.0 ± 0.5 mg/g in each adsorption cycle, which was similar to the result (30.5 mg/g) of batch kinetic studies calculated by the pseudo-second-order model. The breakthrough time and exhaustion time were maintained at about 210 and 266 min, respectively. As such, the slow breakthrough indicates that PEI-chitin is promising as a biosorbent for Pd(II) recovery in industrial applications. The Pd(II) desorption curves obtained through the fixed-bed column experiment are displayed in Figure 6b. The maximum Pd(II) concentration in the effluent was 441.5 ± 29.7 mg/L, and the concentration factor was 8.4 ± 0.8. The removal rate kept at about 64.5% in all three adsorption–desorption cycles. The desorption efficiency in the first, second, and third cycles was 109.5%, 112.3%, and 110.2%, respectively. The high positive error is attributed to the influence of thiourea on the measurement process which has been proven by Sadowska et al. [36]. Taking into account that the adsorption capacity remained stable during the three adsorption cycles, the desorption efficiency may have reached nearly 100%; therefore, PEI-chitin is a biosorbent with good industrial application prospects because it has shown excellent adsorption performance and reusability in fixed-bed column experiments.

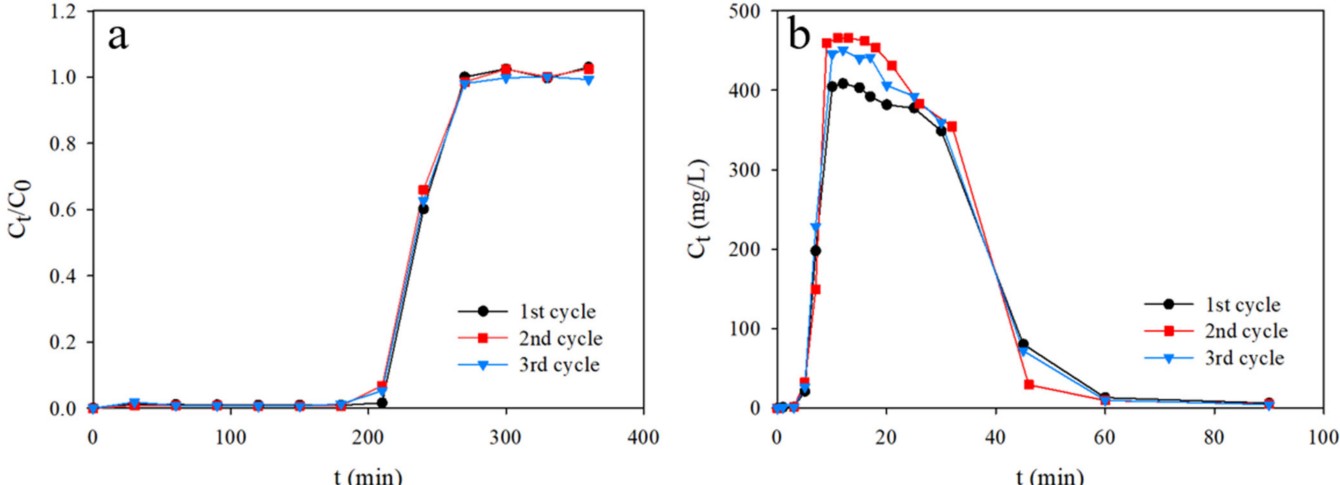

**Figure 6.** (**a**) Adsorption and (**b**) desorption curves of PEI-chitin for Pd(II).

## 4. Conclusions

In this research, the adsorption property of chitin was greatly improved by crosslinking with PEI. FE-SEM analysis showed that raw chitin has a layered and slot-like pore structure, and there was no obvious change after the crosslinking reaction. The elemental analysis confirmed that a small amount of PEI was crosslinked on the surface of raw chitin. $N_2$ adsorption–desorption experiments revealed that, compared with raw chitin, the pore volume and specific surface area of PEI-chitin decreased, while the average pore diameter increased. The Langmuir and pseudo-second-order models were more suitable for clarifying the adsorption performance of PEI-chitin for Pd(II). The calculated maximum Pd(II) uptake of PEI-chitin (57.1 mg/g) was 20 times higher than that of raw chitin (2.8 mg/g). The adsorption equilibrium was reached within 30 min at all studied initial Pd(II) concentrations. The reusability of PEI-chitin was confirmed through three repeated adsorption–desorption cycles in fixed-bed column experiments. Based on all these superior properties, it can be concluded that PEI-chitin can be a cost-effective, reusable, and eco-friendly biosorbent for efficient recovery/removal of Pd(II) from acidic solutions.

**Author Contributions:** Conceptualization, Z.W. and S.W.W.; methodology, Z.W. and S.B.K.; investigation, Z.W.; writing—original draft preparation, Z.W.; writing—review and editing, S.B.K. and S.W.W.; supervision, S.W.W.; project administration, S.W.W.; funding acquisition, S.W.W. All authors have read and agreed to the published version of the manuscript.

**Funding:** This research was funded by the National Research Foundation of Republic of Korea (NRF), grant funded by the Ministry of Science and ICT of Republic of Korea (MSIT), Grant No. 2020R1F1A1065937.

**Institutional Review Board Statement:** Not applicable.

**Informed Consent Statement:** Not applicable.

**Data Availability Statement:** The data set presented in this study is available in this article.

**Acknowledgments:** The authors would like to thank for the support provided by the Joint Laboratory of Gyeongsang National University for sample characterization.

**Conflicts of Interest:** The authors declare no conflict of interest.

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
