# Peer review of "Recovery of Pd(II) from Aqueous Solution by Polyethylenimine-Crosslinked Chitin Biosorbent"

_coatings, doi:10.3390/coatings11050593_

Round 1

Reviewer 1 Report

The authors reported a low-cost biosorbent for precious metal adsorption from acidic solution. The adsorption capacity of the biosorbent achieved dramatic improvement after PEI modification. In addition, the biosorbent showed perfect regeneration and reusability, suggesting its potential in practical applications. This work is quite integrated and has a certain value of novelty, which indicates its promising guidance in the environment and biosorbent material field. This manuscript is well organized and the results are reasonable. I thus recommend acceptance of this manuscript after a few changes as described below.

  1. In Figure 1, the magnification of (a) and (c) are too close to (b) and (c). So (a) and (c) hadn’t provided effective information. Please replace (a) and (c) by images with smaller magnification.
  2. The authors only provide 6 data points in adsorption isotherm curve in Figure 3. It’s not sufficient to analyze the adsorption isotherm. Please provided 2 more points at the high concentration area, and update the Isotherm parameters in Table 3.

Author Response

Thanks for the reviewer's effort on reviewing our manuscript. The detailed responses are listed in the attached file. 

We appreciate for Reviewer's warm work earnestly and hope that the correction will meet with approval.

Once again, thank you very much for your comments and suggestions.

Reviewer 2 Report

Dear Authors,

The manuscript is well written and organised. The topic of research is relevant and conclusion is adequate.

Minor comments:

Introduction, line 50

It seems that the sentence "Because ..." is not complete. Please rephrase

M&M, line 120 - Are the concentration range 50-200 mg/L of Pd in solution realistic ? Please check the concentrations in waste water enriched in this element and add a sentence to justify the selected concentrations

M&M, line 121 -How can you be sure that collection of the supernatant water followed centrifugation is sufficient to guarantee that particles of PEI-chitin are not present? Please provide data to guarantee low loss.
Have you considered the possibility of using a filtration step?

Author Response

(The authors gave the same response as above.)

Reviewer 3 Report

Recommendation: Publish after minor revisions.

Comments:

Sung Wook Won and co-workers report on the recovery of Pd(II) ions from aqueous solutions using modified chitin sorbent. Pd(II) catalysts are widely used in the production of drugs, dyes and other fine chemicals. Thus, the recovery of Pd(II) is widely studied and important for the development of sustainable industrial processes. Sorption on activated carbon is still  most used technology and many researches were conducted to improve this process. The authors propose a novel sorbent based on renewable chitin biopolymer. Their hybrid material was characterized and their sorption properties were carefully investigated. The article is well written and interesting to read. I recommend to publish this article after correction the following minor points:

  1. Page 1, line 12. Please, type 2 as bottom index (N2).
  2. Page 1, line 28. Please, delete point after storage.
  3. Page 1, lines 187-189. BET surface areas are very small (4.24 m2/g and 2.09 m2/g). It means that materials are macroporous (according to SEM and isotherm forms). Nevertheless, the change of meso pores was possible to observe using N2 adsorption isotherms. Please, add this discussion. I recommend also ra evision of the conclusions dealing with this part (BET analysis revealed that compared with raw chitin, the pore volume and specific  surface area of PEI-chitin decreased, while the average pore diameter increased). The changes of macropores cannot be seen using BET analysis and the part of the meso pores is negligible in these materials.

Author Response

(The authors gave the same response as above.)
